# The Potential of Methocinnamox as a Future Treatment for Opioid Use Disorder: A Narrative Review

**DOI:** 10.3390/pharmacy10030048

**Published:** 2022-04-19

**Authors:** Colleen G. Jordan, Amy L. Kennalley, Alivia L. Roberts, Kaitlyn M. Nemes, Tenzing Dolma, Brian J. Piper

**Affiliations:** 1Department of Medical Education, Geisinger Commonwealth School of Medicine, Scranton, PA 18509, USA; akennalley@som.geisinger.edu (A.L.K.); aroberts02@som.geisinger.edu (A.L.R.); knemes@som.geisinger.edu (K.M.N.); tdolma@som.geisinger.edu (T.D.); 2Center for Pharmacy Innovation and Outcomes, Forty Fort, PA 18704, USA

**Keywords:** methocinnamox, opioid use disorder, naloxone, naltrexone, methadone, buprenorphine, overdose, treatment, receptors, addiction

## Abstract

The opioid epidemic is an ongoing public health crisis, and the United States health system is overwhelmed with increasing numbers of opioid-related overdoses. Methocinnamox (MCAM) is a novel mu opioid receptor antagonist with an extended duration of action. MCAM has potential to reduce the burden of the opioid epidemic by being used as an overdose rescue treatment and a long-term treatment for opioid use disorder (OUD). The currently available treatments for OUD include naloxone, naltrexone, and methadone. These treatments have certain limitations, which include short duration of action, patient non-compliance, and diversion. A narrative review was conducted using PubMed and Google Scholar databases covering the history of the opioid epidemic, pain receptors, current OUD treatments and the novel drug MCAM. MCAM could potentially be used as both a rescue and long-term treatment for opioid misuse. This is due to its pseudo-irreversible antagonism of the mu opioid receptor, abnormally long duration of action of nearly two weeks, and the possibility of using kappa or delta opioid receptor agonists for pain management during OUD treatment. MCAM’s novel pharmacokinetic and pharmacodynamic properties open a new avenue for treating opioid misuse.

## 1. Introduction

Over one million Americans have died from overdoses during the opioid epidemic [1]. Opioid addiction and misuse remain a prevalent issue in the United States (US), leading to millions of deaths [2]. Opioids were originally discovered from poppy plants and were used to reduce pain sensation ranging from acute to severe, but in recent history, accessibility to opioids increased across the globe for illicit recreational use, despite increased restrictions for distribution in clinical pain relief therapy [3,4]. The intended use of opioids was for the reduction of pain sensation by agonizing the opioid receptors located in the central nervous system (CNS) [5]. There are three major opioid receptor types, mu (MOR), delta (DOR), and kappa (KOR), but the MOR is the main receptor for providing analgesic effects [5,6]. In the past, many people turned to opioids to relieve daily suffering from chronic pain, and these drugs readily became addictive and created dependence [2]. The ubiquitous use of opioids and the addiction to these drugs in the US has exacerbated the strain on resources in hospitals, emergency rooms, and on first responders as they try to save lives with the limited resources currently available [2]. Naloxone is the only drug available to treat opioid overdose to be approved by the US Food and Drug Administration (FDA) in the last 50 years, while current opioid users are younger and experimenting with synthetic opioids beyond pain relief [7,8]. Naloxone is a competitive MOR antagonist with a high affinity for the MOR receptor used to reverse respiratory and CNS depression in those experiencing an opioid overdose [9,10]. Naloxone does not help decrease future use of prescription or illicit opioids, and the use of synthetic opioids will require higher doses of naloxone, which could increase adverse effects such as tachycardia and hypertension [11,12]. Naloxone is also currently misused at “Narcan parties,” where attendees intentionally overdose knowing they can be rescued by naloxone [13]. Naltrexone has extended-release formulas intended to reduce relapse and promote adherence, yet patient noncompliance and retention continue to be limiting factors [14]. Methadone is commonly used to treat opioid addiction as a replacement for illicit opiates but is itself an addictive substance that can lead to withdrawal if dosage is not closely monitored by a licensed professional [15]. Buprenorphine is currently used to treat OUD, and while it reduces illicit drug use, it is equal to or even less effective than methadone for retaining patients in treatment [15]. Additionally, buprenorphine is sold on the black market on websites such as streetrx.com for those attempting to treat opioid dependency on their own, and it was involved in more drug arrests than methadone, as reported to the Maine Diversion Alert program [16,17]. Methadone and buprenorphine are substitution treatments that substantially reduce opioid deaths during treatment, but immediately following cessation of treatment, the mortality risk notably increases [18]. For these reasons [1,2], there is an urgent need for new opioid misuse interventions [19].

Methocinnamox (MCAM) is a novel drug candidate that is a pseudo-irreversible MOR antagonist, thereby preventing other opioid agonists from binding for a two-week period [19,20,21]. Due to the long-lasting effects of MCAM, it can be a safer and more effective alternative medication for OUD [22,23]. MCAM has the potential to change the course of opioid misuse and help prevent subsequent renarcotization after administration [24,25]. Although not yet tested in human trials, MCAM’s currently understood safety profile exhibits minimal side effects. This brief review will explore how MCAM’s unique function could be useful in reducing the substantial personal and societal burden of the opioid crisis.

## 2. Main Body

### 2.1. Methods

A narrative review was conducted using PubMed and Google Scholar databases utilizing the following key terms: methocinnamox, MCAM, naloxone, naltrexone, buprenorphine, buprenorphine–naloxone, methadone, opioid overdose, opioid crisis, opioid abuse, mu-receptor, kappa-receptor, delta-receptor, inverse agonist, and naloxone and placebo. These terms and pharmacotherapies were included due to their relevance to the opioid epidemic, opioid overdose, and MCAM. Chemical structure was constructed with ChemDraw version 19.0.0. No date range or journal exclusion criteria were applied.

An initial review of the literature resulted in 2574 articles. Included papers discussed: history of the opioid epidemic in the United States; relevant receptors and their significance; current treatments (methadone, buprenorphine, naloxone, and/or naltrexone); and MCAM. Once sufficient detail was obtained, no further papers were included for all sections; however, all published articles with MCAM as the primary focus were included. The remaining 93 articles referenced within this paper were organized into the following topics: 21 detailed the history of the opioid epidemic; 28 provided information on the opiate pain receptors; 32 described the currently available OUD treatments; 17 focused on MCAM.

### 2.2. Opioid Epidemic

In the 1800s, opiates were widely marketed as a safe and effective form of pain alleviation [26]. Consequently, the absence of federal regulation on frequent opioid distribution and use drew widespread concern, which eventually led to the enactment of the 1914 Harrison Narcotic Control Act [27]. While this prompted nationwide stigmatization of opioid use for non-cancer chronic pain management, it was later followed by a drastic shift in 1995 in public attitude that advocated for the recognition of pain as a “fifth vital sign” [28]. As a result, several entities such as the Institute of Medicine, the Federation of State Medical Boards, and the Drug Enforcement Agency pushed for fewer regulations over opioid prescriptions, thereby encouraging healthcare providers to provide increased pain relief for patients. Additionally, the FDA approved an extended-release oxycodone formulation in 1995 as a safer opioid alternative to the immediate-release version because of its slow and sustained release of the medication [29]. Pressure by pharmaceutical companies, patients, and federal funding requirements further contributed to the overaggressive prescription of opioid analgesics that ultimately led to an iatrogenic opioid epidemic in the US [30,31]. The Centers for Disease Control reported that in 2016, more than 42,000 Americans died from an opioid overdose, marking a 27% increase from the previous year [32]. In 2017, the rate increased by 45.2%, indicating an increased prevalence of opioid misuse [33]. More than 11.5 million Americans misused opioids, and roughly 2.1 million were formally diagnosed with an OUD [34]. Later that year, the US Department of Health and Human Services declared the opioid epidemic as a public health emergency. Despite concerted efforts by medical practitioners to reduce opioid prescriptions and increased availability of buprenorphine, methadone, and naloxone [35,36,37], opioid overdose continues to rise with one hundred thousand cases annually [38], with the illegal manufacturing of fentanyl and its analogs as the leading cause [39]. This can be attributed to fentanyl’s high potency, with a strength 30–50 times greater than heroin, its rapid onset of action, long duration of desired effect, and low production costs. While the rate of heroin-related overdose deaths has started to stabilize, from 2013–2016, synthetic opioid-related deaths, such as those caused by fentanyl and its analogs, increased by 88% [30,40].

Public health experts agree that tackling the opioid epidemic will require interdisciplinary collaboration between medical providers, social service agencies, federal regulation, and community support [8,41,42]. The lingering effects of the epidemic are rampant in low-income communities, predominantly African American and Hispanic communities, and are currently exacerbated by COVID-19’s social and health impacts [43,44]. The pandemic increased isolation, general stress, grief, and financial instability, all of which are known to trigger addiction and relapse [45]. Overdose-related deaths rose due to the pandemic; with mandated isolation, users had no one nearby to administer rescue medication upon overdosing. Additionally, the inability to access suppliers caused individuals to experience a decrease in tolerance; thus, when able to use opioids again as mandates lifted, a previously safe dose became fatal [45]. Potential proposed solutions include increasing harm reduction programs, educating medical providers on safe opioid prescribing, eliminating stigma around OUDs, as well as finding safer alternatives to pain management [3,42,46]. Current OUD therapeutics include methadone, buprenorphine, naloxone, and extended-release naltrexone, all of which function by reducing opioid withdrawal symptoms and cravings [5,47]. However, drug addiction is generally recognized as a complex biopsychosocial condition. These medications can only successfully resolve the opioid crisis by working in tandem with public health efforts that include both prevention and harm-reduction approaches [48]. MCAM presents potential for a new avenue of OUD treatment.

### 2.3. Pain Receptors

MCAM is a long-lasting, pseudo-irreversible (non-covalent binding), potent, MOR antagonist that reversibly binds KOR and DOR and has no known interaction with other nociceptors. Thus, KOR and DOR agonists could be provided concomitantly for pain relief during treatment for OUD, although KOR agonists are known to cause dysphoria in humans and therefore may not be useful for pain relief therapy [20,22,23,24,49,50,51]. The unique pharmacodynamics of MCAM contribute to its long-lasting effects. The need for new MOR to induce the euphoric and depressive effects of opioid receptor agonists as receptor turnover is what limits the duration of action (DOA) [19,24]. This is crucial because MOR agonists can potentially not only induce the G protein-coupled receptor (GPCR) pathway, but can also induce β-arrestin activation, leading to adverse effects such as respiratory depression [52,53]. MOR, KOR and DOR belong to the largest membrane receptor family called the trimeric GPCR superfamily. Opioids activate the inhibitory (Gi) signaling pathway to initiate analgesic functions [54,55,56,57]. GPCRs are known for their trimeric subunits consisting of alpha (Gα), beta (Gβ), and gamma (Gγ) [58]. After an opioid agonist (endogenous or exogenous) binds, a signal stimulates Gα to migrate and suppress adenylate cyclase activity, thereby reducing cyclic AMP production [58]. The Gβγ acts as a modulator for the signaling pathway, resulting in reduced neurotransmitter release and membrane hyperpolarization [58].

Since GPCRs are widespread, these are the target for 50% of marketed pharmacological therapeutics, revolving around the common amino-terminal peptide sequence, Tyr-Gly-Gly-Phe, which is referred to as the “opioid motif”, as it directly interacts with the opioid receptor [59]. Examples of MOR agonists include oxycodone, fentanyl, heroin, morphine, and methadone. Buprenorphine is a partial MOR agonist but a KOR antagonist [60,61]. MOR antagonists include naloxone, naltrexone, and MCAM. Activation of the KOR hyperpolarizes neurons that are activated indirectly by MOR, thus creating synergistic antinociception [7,62]. Pain is multidimensional and dependent on subjective thresholds. Chronic pain, which may be concurrent with anxiety, may be associated with neuroplastic changes in the amygdala, which may heighten the emotional and affective consequences of pain [63,64]. Opioid analgesics are highly effective in most cases against acute pain, but the desired effects mediated by the opioid receptor family may lead to craving, addiction, or dependence as a result of neurological changes [65,66,67,68]. Repetitive opioid use will thus increase the threshold for analgesic effects secondary to compensatory upregulation of vesicular calcium content while developing opioid tolerance, and it may decrease one’s quality of life [55,69,70].

### 2.4. Current OUD Treatments

The current pharmacological treatments for opioid overdose and misuse are administration of methadone, buprenorphine, naloxone, and naltrexone [9,10,12,71,72,73]. Methadone and buprenorphine are MOR agonists that may prevent withdrawal symptoms in those recovering from OUD but pose risks for opioid overdose, particularly when combined with other substances [71,74,75]. However, naltrexone and naloxone are opioid antagonists, the latter being the only emergency rescue for opioid overdose and opioid induced respiratory symptoms [10,76]. Naltrexone is used to treat OUD and opioid dependency, usually post-opioid cessation, whereas naloxone can be used concomitantly with prescribed opioids such as buprenorphine [71,72,73,77]. Other pharmacological uses have been identified for naloxone and naltrexone such as treatment for alcohol dependence, and possible treatments for internet sex addiction and Hailey–Hailey disease [78,79,80]. Unfortunately, these medications were ineffective for smoking cessation [81].

Administration of methadone or buprenorphine significantly reduced opioid-related deaths caused by nonfatal opioid overdose over a 12-month follow-up period by 59% and 38%, respectively [82]. The abrupt discontinuation of opioids does not show great success rates and may result in relapse [8,68]. Use of these drugs in conjunction with psychosocial therapy are the best for treatment success in those with OUD [8,71,73]. While both methadone and buprenorphine are synthetic opioids and are used in medication-assisted treatment (MAT) of OUD, they possess different mechanisms of action in terms of complete versus partial MOR agonism and adverse drug reactions (ADRs). Methadone is a long-acting full agonist that binds the MOR, preventing withdrawal symptoms such as nausea and vomiting for at least 24 h, while conferring analgesia and reducing opioid cravings [83,84]. Conversely, buprenorphine is a partial agonist at the MOR, making it less potent than methadone with decreased ADRs [84,85]. Buprenorphine and its metabolite norbuprenorphine function as a mixed KOR antagonist and DOR agonist [61,85]. Methadone has restricted availability in the US and requires daily dosing, raising the issue of patient compliance. Close monitoring of dosing for detoxification by a practitioner for each individual is an additional healthcare burden [86]. Buprenorphine, even at high doses, is less effective for patient retention than methadone [15]. Buprenorphine’s purported “ceiling effect” reduces the risk of misuse or overdose by preventing an increase in opioid effects, or euphoria, beyond a designated threshold [84]. However, buprenorphine may precipitate withdrawal, a condition that occurs without an adequate detoxification period from opioid drugs, due to its high affinity for the MOR [84,85]. The combination of buprenorphine with benzodiazepines causes respiratory depression [84,85].

An additional drug that is used in MAT programs is naloxone, which is often paired with buprenorphine in an oral tablet form to prevent strong withdrawal symptoms and block the euphoric effects induced by other opioids [72]. The action of naloxone is by competitive binding to the MOR as a high affinity antagonist, and some neurochemists believe it acts as an inverse agonist [10,87]. Administration of naloxone intravenously, intramuscularly, subcutaneously, intranasally, and even inhalation through the endotracheal tube for intubated patients during an opioid overdose competitively binds the opioid receptors to reverse respiratory and CNS depression [88]. After a 13 μg/kg dose of naloxone, 50% of the receptors in the brain were occupied, but due to the rapid association and successive dissociation of naloxone from the receptors, toxicity reversal may be insufficient, and the patient may experience renarcotization requiring subsequent doses [12,88,89]. Although regarded as exceedingly safe, ADRs for naloxone can include tachycardia, hypertension, gastrointestinal upset, hyperthermia, cravings, nausea, vomiting, and, rarely, severe cardiovascular events [9,12,88]. Naloxone also blocks the descending pain control system, thus diminishing the placebo pathway for pain perception by interfering with the coupling between the rostral anterior cingulate cortex and the periaqueductal gray [90,91]. Although not the first-line treatment for opioid overdose, naltrexone is used to reduce opioid misuse in those with OUD [73,76].

Naltrexone is commonly characterized as an opioid receptor antagonist, although some suggest an inverse agonist function based on its intracellular signaling properties [87]. Naltrexone is prescribed to reduce opioid use in those attempting to practice abstinence from opioids but suffer from OUD [73,76]. Interventions for opioid misuse involving naltrexone, rather than receptor agonists such as buprenorphine, have been successful when paired with behavior intervention and are a promising alternative treatment for opioid misuse in pregnant women [92,93]. For opioid dependent pregnancies, naltrexone had reduced opioid misuse in mothers and significantly decreased neonatal abstinence syndrome in infants when compared to buprenorphine [92]. In contrast to naloxone’s associated acute withdrawal symptoms, naltrexone reduces symptoms of withdrawal for patients and even lowers the risk for overdose with the use of buprenorphine as an OUD treatment with no significant ADRs [77]. While also having a favorable safety profile similar to naloxone, the potential adverse effects of naltrexone include mild to moderate injection site reaction, nausea, and gastrointestinal upset [94]. See Table 1 for a summary comparing these OUD pharmacotherapies. With the intervention’s limitations outlined above, MCAM may prove beneficial as a treatment to combat the opioid crisis if human trials produce results consistent with animal trials.

### 2.5. Methocinnamox

MCAM, shown in Figure 1, was first mentioned in a publication in 2000 by researchers from the University of Michigan Medical School and the University of Bristol, but was initially discarded because it was believed to be useful only for MOR research [8,108]. However, it is currently being studied for its promise in the opioid crisis as a long-term OUD treatment [109,110]. In animal models, a single subcutaneous dose of MCAM rescues a subject from acute opioid overdose and prevents subsequent overdose for up to two weeks with minimal adverse effects [21,22,104,108,111]. Currently, the only known possible adverse effect for MCAM in non-dependent individuals was hyperventilation upon rescue [104]. One study noted a slightly increased response to food (a non-drug alternative) several days following a single injection due to MCAM blocking opioids from binding [22]. MCAM itself does not stimulate hunger, it removes the decrease in appetite, a side effect of opioids, by disallowing opioid binding. Some studies predict no potential ADRs when combined with benzodiazepines and alcohol, meaning that MCAM could be a preferred pharmacotherapy for individuals co-using alcohol or benzodiazepines [19,22]. MCAM has not been shown to alter responses to food, heart rate, blood pressure, body temperature, or social and physical activity and no indication of developing tolerance nor physical dependence [19,22]. MCAM is a potent MOR antagonist and shows no agonistic effects, even at high concentrations, with the longest duration and highest potency when injected subcutaneously over other methods of administration [20,49,104].

Naltrexone and naloxone injections become ineffective in less than a single day, with durations of action lasting 1–2 h [88,89]. A single injection of MCAM has a duration of action of thirteen days, reaching peak concentration 15–45 min after injection with a half-life of roughly 70 min [19,24,105]. Currently, MCAM’s pharmacokinetic properties are not fully understood, but the effectiveness at low plasma levels suggests that pharmacodynamic properties, rather than pharmacokinetic factors, play a prominent role in its long-lasting effects [19]. The evidence for pseudo-irreversible binding includes its non-reversible, insurmountable, and time-dependent antagonism of MORs [106]. A recent report with human embryonic kidney (HEK) cells expressing human opioid receptors showed that in addition to pseudo-irreversible orthosteric antagonism of MORs directly blocking binding, MCAM also utilized allosteric antagonism at an unknown site at a lower affinity, which alters ligand affinity and/or intrinsic efficacy of MOR agonists [106]. MCAM contains a weak Michael acceptor group that non-covalently binds to the MOR orthosteric site [106]. This suggests no true irreversible binding despite its seemingly irreversible effects. The pseudo-irreversible binding effectively incapacitates MORs; thus, cells need to synthesize nascent MORs to reestablish previous functionality [106]. Due to this, MCAM possesses a uniquely long DOA. As for MCAM’s interaction with DOR and KOR, the MOA was consistent in the opioid receptor expressing HEK cells, and in vivo, with simple competitive antagonism [106].

Repeated administration of MCAM every twelve days in rodents remained effective for over two months without altering the duration of opioid withdrawal with no major ADRs and no decrease in effectiveness, suggesting positive potential for long-term OUD treatment [19,21,99,113]. Naltrexone, naloxone, and MCAM are effective for acute reversal and prevention of respiratory depression and other overdose symptoms due to their effects on opioid receptors, but only MCAM prevents renarcotization in the hours and days following emergency intervention [19,48,104,105,114,115]. Naltrexone and naloxone bind competitively, meaning higher amounts of an agonist will overcome their intended effects, requiring a higher dose of either therapy to reverse initial and subsequent overdoses post-antagonist injection [105]. MCAM binds non-competitively, making it insurmountable and therefore more effective at blocking effects of opioids in the short- and long-term [19,24,106,114]. Additionally, MCAM is naloxone-insensitive with no notable drug interactions, meaning that there is a possibility that the two drugs could be administered concurrently for immediate rescue and prevent subsequent renarcotization [106]. With over-the-counter availability of naloxone, overdose-related deaths have decreased, but subsequent renarcotization and therefore consequent overdoses leading to death remains an issue [88,106]. If a shorter acting formulation of MCAM was combined with naloxone, renarcotization risk could significantly decrease and potentially further reduce opioid overdose-related deaths without inducing withdrawal. Naloxone would provide immediate rescue, and MCAM would prevent subsequent renarcotization. An extended-release or longer acting naloxone formulation retains the risk of renarcotization due to being surmountable [106]. Because the DOA differs between subcutaneous and intravenous methods of administration [104], it may be plausible that a different administration method could have a short enough duration to prevent renarcotization, but losing its effect before withdrawal is precipitated.

MCAM can act as a preventative therapy for opioid misuse, indicating possible use at discharge from treatment facilities following a detoxification period, as well as use during ongoing therapeutic intervention [22,24,48,114]. Its prolonged DOA results in a long period before needing the next dose, which is hypothesized to relatively prevent patient noncompliance that is seen with extended-release naltrexone for outpatient treatment, including eliminating the possibility of an individual removing an implant [22,24]. In cases where effects lasting roughly five days or less are needed, such as preventing renarcotization in the hours and few days following an overdose but not for long-term treatment of OUD, intravenous administration of MCAM would be preferable because this method’s DOA is roughly 5 days [104,105]. There is discussion of creating an oral pill form of MCAM, an extended-release form, and faster acting intranasal and intramuscular formulations, but further study of the drug is needed before these will be developed [19,22,105]. MCAM also blocks the physiological and behavioral effects of MOR agonists such as unfavorable impacts of sensitivity to mechanical stimulation, gastrointestinal motility, and appetite [21,22,25,107]. MCAM does not impact memory and other cognition [112]. For these reasons, the adverse effect profile is encouraging. However, it is important to recognize that no testing has been conducted in humans [108]. It is currently unknown if long-term blockade of the MOR would attenuate endorphin and enkephalin signaling sufficiently to alter mood. Given the widespread impact of opioid overdoses [1,2], novel strategies are desperately needed.

## 3. Conclusions

The increased prevalence of OUD cases and opioid-related deaths are an ongoing public health crisis in the US. While opioid antagonists, naltrexone and naloxone, are essential drugs used to treat OUD and reverse the effects of an overdose, respectively, they have risks that pose considerable limitations to their efficacy. These risks include withdrawal, poor patient compliance, short durations of action, lack of concurrent antinociceptive treatment, ability to surmount opioid receptor blockade, and potentially dangerous drug–drug interactions, especially for those with comorbid addictions [12,19,22,88,94]. Opioid agonists methadone and buprenorphine present their own limitations in OUD treatment such as dependence, restricted availability, poor patient retention, patient noncompliance, drug–drug interactions, necessary detoxification, and potential withdrawal [15,84,85,86]. Buprenorphine is a Schedule III and methadone is a Schedule II substance in the US; thus, misuse and diversion of these substances is an ongoing challenge [16,17]. The demand for novel therapeutics to decrease the misuse and overuse of opioid drugs and resulting overdoses provides an opportunity for MCAM to make a positive impact. By retaining the safety benefits of naltrexone and naloxone and providing a longer DOA with a novel mechanism, MCAM is a promising pharmacological addition. Using non-MOR agonists such as KOR or DOR agonists concomitantly with MCAM also presents a potential intervention method, allowing for antinociceptive effects during the withdrawal process and OUD treatment [48,101,115,116]. The preclinical phase of MCAM drug development began in 2005 with testing in mice, rats, and non-human primates [21,48,104,105,106,107,110,111,114]. Researchers aim to begin phase I clinical trials by 2022, although the COVID-19 pandemic may have delayed trials [108,111,117]. Future research should focus on comparing MCAM to currently available therapies in vivo. MCAM has the potential to transform the future of OUD treatment, thereby reducing the healthcare and societal burdens caused by the opioid epidemic [1,2] and improving the lives of millions.

## Figures and Tables

**Figure 1 pharmacy-10-00048-f001:**
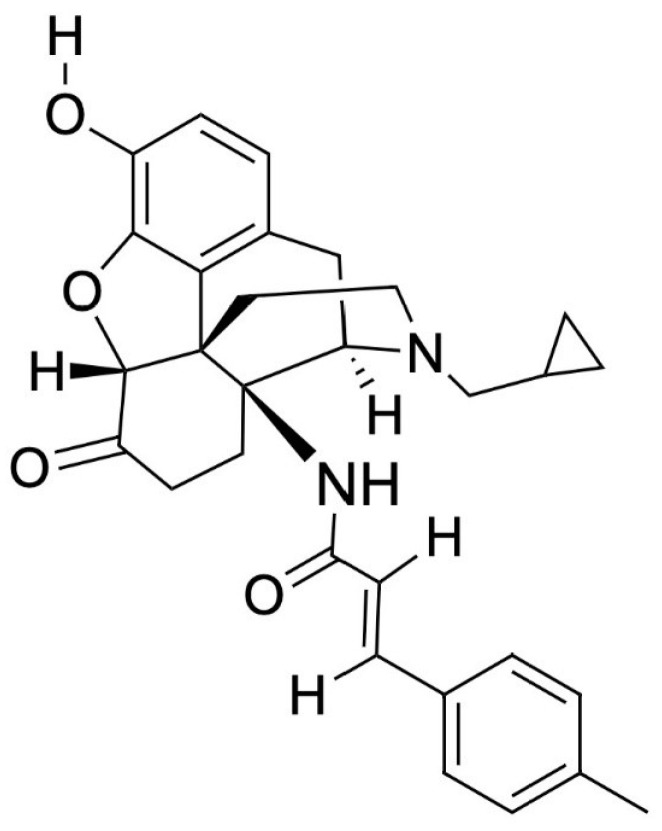
Methocinnamox’s chemical structure. Molecular Formula: C_30_H_32_N_2_O_4_, PubChem CID: 46877713, IUPAC name: (E)-N-(4R,4aS,7aR,12bR)-3-(cyclopropylmethyl)-9-hydroxy-7-oxo-2,4,5,6,7a,13-hexahydro-1H-4,12-methanobenzofuro3,2-eisoquinolin-4a-yl-3-(4-methylphenyl)prop-2-enamide (compound/methocinnamox) [112].

**Table 1 pharmacy-10-00048-t001:** Comparison of opioid use disorder and opioid overdose treatment drugs. Table includes naloxone [95,96], naltrexone [97,98], methadone [99,100,101], buprenorphine [102,103], and methocinnamox (MCAM) [19,24,104,105,106,107]. DOA: duration of action, μ: mu, κ: kappa, δ: delta opiate receptor, min: minutes, h: hours.

Treatment Drug	Method of Administration	Mechanism of Action	Onset of Action	Duration of Action	Strengths	Limitations
naloxone	Intravenous Intramuscular Subcutaneous Intranasal Inhalation	Reversible μ, κ and δ competitive antagonist	1–5 min	1–2 h	Rescue from overdose, wide therapeutic window	Short DOA, community misuse, risk of renarcotization, precipitates withdrawal, drug-drug interactions, surmountable
naltrexone	Intravenous Intramuscular Subcutaneous ^1^ Oral	Reversible μ, κ and δ competitive antagonist	15–30 min	>72 h	Use during pregnancy, extended-release formula	May precipitate withdrawal, patient noncompliance, drug-drug interactions
methadone	Intravenous Intramuscular Subcutaneous Oral	μ and δ agonist	30–60 min	4–8 h, single dose; 22–24 h, continuous dosing	Prevents withdrawal, reduces opioid cravings, pain relief	Patient noncompliance, dependence, misuse and diversion, restricted availability in US, close monitoring of dosage, many drug-drug interactions, ADRs
buprenorphine	Intravenous Intramuscular Subcutaneous Oral Buccal Sublingual Transdermal	Partial μ agonist, κ, δ competitive antagonist	10–30 min	2–24 h	Prevents withdrawal, use during pregnancy, prevents euphoria and overdose symptoms	Less potent than methadone, patient retention inferior to methadone, may precipitate withdrawal, drug-drug interactions, ADRs
MCAM	Intravenous Subcutaneous	Pseudo-irreversible μ, not competitive antagonist ^2^, reversible κ, δ competitive antagonism	15–45 min	5 days to 2 weeks	Long DOA, not surmountable, prevents renarcotization, lacks notable drug–drug interactions, antinociceptive concomitant treatment possible	Precipitates withdrawal, not yet tested in humans

^1^ Subcutaneous formula is a pellet implant. ^2^ MOA incompletely understood.

## Data Availability

Not applicable.

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
