# Peer review of "The Potential of Methocinnamox as a Future Treatment for Opioid Use Disorder: A Narrative Review"

_pharmacy, 2022, doi:10.3390/pharmacy10030048_

Round 1

Reviewer 2 Report

Thank you for inviting me to review this article. It is of high interest, relevance, and is time-sensitive to the field of addiction medicine. Overall, this is a promising manuscript; however, it has points that need to be addressed before it can be published.

The format needs to adhere to a narrative, systematic, or scoping review so that it can be clearly interpreted and useful for other researchers. There are a number of published examples in PubMed. (Two samples are listed below.)

Maggio LA, Larsen K, Thomas A, Costello JA, Artino AR Jr. Scoping reviews in medical education: A scoping review. Med Educ. 2021 Jun;55(6):689-700. doi: 10.1111/medu.14431.

Gasparyan AY, Ayvazyan L, Blackmore H, Kitas GD. Writing a narrative biomedical review: considerations for authors, peer reviewers, and editors. Rheumatol Int. 2011 Nov;31(11):1409-17.

This does not need to be exhaustive, but you must have a results section which lists the published literature that can be quickly scanned including a table or appendix. For example, what are the negative effects of MCAM? The authors have presented only positive examples. A balanced approach should be presented to prevent bias.  

A few minor comments:

Line 221-22, “With the intervention’s limitations outlined above, MCAM may prove beneficial as a treatment to combat the opioid crisis.”

This is an awkward place to put this sentence, as there are no human trials to date.

Line 238-39. “…one study noted a slightly increased response to food (a non-drug alternative) several days following a single injection”

Please clarify.

“MCAM has not been shown to cause a decrease in 241 response to food…” Again, please clarify if these are changes in appetite.

cAMP, needs to be explained.

I am confused by lines 283-85: how does MCAM prevent the effects of withdrawal? Please clarify.

MOR, KOR, and multiple others need to be explained and spelled out before being used as acronyms.

Reviewer 3 Report

Overall, this paper was well written with interesting findings. Though I think there were areas of improvement. For example, since this is a review paper, instead of following standard format (intro, method, results, discussion, conclusion) without result section, you can change the format to 'intro', 'body', 'conclusion'. Also, considering no human studies conducted, I think the outcomes from animal studies can be more explicitly included. Lastly, please ensure to do proofreading since I found some grammar errors and address all comments in the attachment.

Round 2

Reviewer 2 Report

Thank you for your edits and additions. This paper is greatly improved but still requires editing for Grammar and Language Use. Please see the following suggestions, which should take minimal effort.

From the Abstract:

  1. The Potential of Methocinnamox as a Future Opioid Use Disorder Treatment: A Narrative Review

Should have person-first language: “…a Future Treatment for Opioid Use Disorder”

  1. “The MCAM has potential”; remove “The”

  1. “A literature review was conducted using…” Is it a literature review or a narrative review? This is also mentioned in the Methods. Be consistent with your naming of the type of review.

  1. The Opioid Epidemic section “used” 25 articles. This is awkward wording and does not properly convey the methodology.

Try restating this…one example would be “An initial review of the literature resulted in # articles.” [Then describe any further screening process if any.] The remaining # were organized into the following topics: 28 focused on pain receptors; # considered [subject]; ….”

  1. “Additional papers were used in the introduction and conclusion sections”. THIS is not part of your methodology for how you wrote the article. Delete this from this section. It is expected you would gather background material.

  1. “Additionally, inability to access suppliers caused users to experience a decrease in tolerance,”

REPLACE the word “users” with “individuals”. I recognize that StatNews.com utilizes this word, but the Addiction Medicine Community advocates for person-affirming language.

Line 282: “…meaning MCAM could be a preferred pharmacotherapy for individuals co-abusing alcohol”

                        REPLACE “co-abusing” with “co-using”

https://www.asam.org/docs/default-source/default-document-library/nidamed_wordsmatter3_508.pdf?sfvrsn=5cf550c2_2

OPTIONAL: I would recommend citing peer-reviewed sources for your peer-reviewed articles as opposed to a news article.
